# Circulating microRNAs from the Molecular Mechanisms to Clinical Biomarkers: A Focus on the Clear Cell Renal Cell Carcinoma

**DOI:** 10.3390/genes12081154

**Published:** 2021-07-28

**Authors:** Claudia Tito, Elena De Falco, Paolo Rosa, Alessia Iaiza, Francesco Fazi, Vincenzo Petrozza, Antonella Calogero

**Affiliations:** 1Department of Anatomical, Histological, Forensic & Orthopedic Sciences, Section of Histology & Medical Embryology, Sapienza University of Rome, Laboratory Affiliated to Istituto Pasteur Italia-Fondazione Cenci Bolognetti, 00161 Rome, Italy; claudia.tito@uniroma1.it (C.T.); alessia.iaiza@uniroma1.it (A.I.); francesco.fazi@uniroma1.it (F.F.); 2Department of Medical-Surgical Sciences and Biotechnologies, Sapienza University of Rome, 04100 Latina, Italy; elena.defalco@uniroma1.it (E.D.F.); p.rosa@uniroma1.it (P.R.); vincenzo.petrozza@uniroma1.it (V.P.); 3Mediterranea Cardiocentro, 80122 Naples, Italy

**Keywords:** circulating miRNAs, biomarkers, ccRCC, liquid biopsies

## Abstract

microRNAs (miRNAs) are emerging as relevant molecules in cancer development and progression. MiRNAs add a post-transcriptional level of control to the regulation of gene expression. The deregulation of miRNA expression results in changing the molecular circuitry in which miRNAs are involved, leading to alterations of cell fate determination. In this review, we describe the miRNAs that are emerging as innovative molecular biomarkers from liquid biopsies, not only for diagnosis, but also for post-surgery management in cancer. We focus our attention on renal cell carcinoma, in particular highlighting the crucial role of circulating miRNAs in clear cell renal cell carcinoma (ccRCC) management. In addition, the functional deregulation of miRNA expression in ccRCC is also discussed, to underline the contribution of miRNAs to ccRCC development and progression, which may be relevant for the identification and design of innovative clinical strategies against this tumor.

## 1. Biogenesis and Function of microRNAs

Genome-wide transcriptional analyses have led to the discovery of many non-coding RNAs (ncRNAs) that are transcribed but do not encode any protein. ncRNAs represent a large fraction of the human genome (up of 70%) and are classified into two groups based on their size: small ncRNAs (<200 nt), such as microRNAs (miRNAs), small interfering RNAs (siRNAs), PIWI-interacting RNAs (piRNAs), transfer RNAs (tRNAs), small nuclear and nucleolar RNAs (snRNAs and snoRNAs), small conditional RNAs (scRNAs) and long ncRNAs longer than 200 nucleotides (>200 nt). MicroRNAs (miRNAs) are a class of small endogenous non-coding RNA of 19–22 nucleotides, evolutionarily conserved and located in intergenic regions and in exonic or intronic sequences of protein coding genes [1,2]. MiRNAs play a role in epigenetic mechanisms [3] and their biogenesis is a specific and highly regulated process occurring in the nucleus and cytoplasm through the action of different proteins and enzymes. First, miRNAs are transcribed in the nucleus by RNA Polymerase II as a sequence of approximately 170 nucleotides (pri-miRNAs) and then cleaved into a 70-nucleotide sequence (pre-miRNAs) by RNA Nuclear RNase III Drosha in association with its cofactor, DiGeorge Syndrome Critical Region gene 8 (DGCR8). Once pre-miRNAs are exported by Exportin5 (XPO5) protein into the cytoplasm, mature small double-stranded miRNA/miRNA duplexes are processed by RNase III endonuclease Dicer [4]. The two strands composing miRNA duplex are indicated as -3p and -5p. One of the two miRNA strands (guide strand) is incorporated into a ribonucleoprotein complex, known as the RNA-induced silencing complex (RISC), where a member of the Argonaute (Ago) family of proteins, Ago1-4, provides a unique platform for target recognition and gene silencing [5,6,7]. The other strand (passenger strand) is more likely to be discarded and degraded [8,9]. The mechanism of miRNA action is determined by their ability to bind to the complementary sequence in the 3′-untranslated region (UTR) of target mRNAs, therefore regulating their expression by reducing mRNA stability and/or inhibiting translation. Binding to 5′-UTR or the coding sequence (CDS) has also been described [10,11]. The complete or partial pairing between miRNA and target mRNA leads to the degradation or inhibition of protein translation genes, respectively [12,13]. In both cases, they negatively affect gene expression emerging as relevant players in many biological processes, such as proliferation, apoptosis, migration, and differentiation [14]. Furthermore, they are involved in cell development and fate determination, viral infection, immune response, angiogenesis, and the development of many pathologies [15,16]. Given their involvement in biological and functional pathways, many studies have reported the dysregulation of miRNA expression in several types of cancer, including breast cancer [17,18], hepatocellular carcinoma [19], lymphoma [20], lung [21], prostate [22] and colorectal carcinoma [23]. On the basis of this evidence, miRNAs can act as both oncogenes (oncomiRs) or tumor suppressor genes (tumor suppressive miRs), regulating genes and pathways correlated with cancer progression. Consequently, the aberrant expression of miRNAs is strongly associated with tumorigenesis [24,25,26,27]. Generally, the alteration in the miRNA expression between tumoral and normal cells, depends on the alteration of their genomic location due to chromosomal abnormalities, epigenetic mechanisms, and mutations in their DNA sequence or defects during biogenesis. As a result, miRNA expression profiling represents one of the main markers of tumor origin, acquiring relevance for diagnosis, prognosis [27,28], and therapeutic treatment [26,27,29]. Several advanced technologies, which include bead-based flow cytometry, northern blot, RNase protection assay, quantitative reverse transcription-polymerase chain reaction (qRT-PCR), and microarray analysis, have contributed to identify a variety of miRNAs differentially expressed in human tumors [12].

## 2. Circulating miRNAs: Free in Biofluids, Secreted in Vesicles, or Bound to a Ribonucleoprotein Complex

MiRNAs can be secreted by cells into biofluids as circulating miRNAs and in cancer tissues, therefore emerging as relevant diagnostic and prognostic cancer biomarkers. MiRNAs can be detected in multiple body fluids, such as saliva, breast milk, tear, bronchial lavage, cerebrospinal fluid, urine, amniotic fluid, blood plasma, and serum. Circulating miRNAs are secreted into the extracellular environment and encapsulated in extracellular vesicles or bound to protein complexes [30]. In both cases, miRNAs are protected from RNase-mediated degradation and remain highly stable in the harsh conditions of several treatments such as boiling, storage in low temperature, pH changes, or freeze-thaw cycles [31]. The extracellular vesicle containing miRNAs can be classified into different types depending on their size, cellular origin, function, and density of lipoproteins [32,33]. Exosomes are small membrane vesicles (diameter of 50-100nm) produced by all eukaryotic cells through the fusion of multivesicular bodies (MVBs) with the plasma membrane [34]. Since exosomes carry specific mRNAs, lipids, proteins, and ncRNAs to recipient target cells at distant sites, they represent a novel mechanism of intracellular communication [35]. Several reports have identified the presence of a great series of miRNAs in exosomes. For example, the study of Skog et al. found a similar distribution of 11 miRNAs between human primary glioblastoma cells and exosomes released into the extracellular environment. These identified miRNAs are involved in cell migration, cell proliferation, and angiogenesis [36]. Another study by Rabinowits et al. observed the same expression of 12 specific miRNAs between exosomes and parental tumor cells in patients with and without lung adenocarcinoma. These miRNAs are overexpressed in non-small-cell lung cancer compared to normal tissues [37]. Valadi et al. detected 121 miRNAs in the exosomes released from human mast cells that can interfere with 24,000 mRNAs. Among them, they identified a series of miRNAs, such as miR-1, miR-17, miR-18, miR-181, and miR-375, which regulate many biological processes, including angiogenesis, hematopoiesis, and exocytosis, favoring tumorigenesis [38]. Collectively, exosomal miRNAs can influence the tumor environment at distant sites of the body, regulating the gene expression, and increasing the tumor growth. Besides exosomes, miRNAs can be packed into both the microvesicles (diameter of 100–1000 nm) and into the apoptotic bodies (diameter of 1000–4000 nm) released by apoptosis [34]. Hunter et al. observed in plasma microvesicles a high expression of miRNAs that regulate hematopoiesis, cellular differentiation, and proteins involved in cell cycle progression [39]. Likewise, the role of apoptotic bodies was demonstrated by Zernecke et al. in atherosclerosis disease. During this process, membranous vesicles are enriched with miR-126, which mediates the production of chemokine CXC motif ligand 12 (CXCL12) and inhibits the expression of its target genes in the recipient cells, reducing the atherosclerosis disease [40].

MiRNAs can also be delivered to the recipient cells by high density-lipoprotein (HDL). In a recent study HDL-miRNA profiles in normal and affected familial hypercholesteremia disorder patients were analyzed. A different expression profile of miRNAs associated with lipoprotein was demonstrated between healthy subjects and patients: hsa-miR-135a∗, hsa-miR-188-5p, and hsa-miR-877 were enriched in healthy subjects; with hsa-miR-223, hsa-miR-105p, and hsa-miR-106a in patients. The authors confirmed that the mechanism of HDL miRNA release is mediated by sphingomyelinase 2 (nSMase2) pathways. Indeed, the inhibition of nSMase2 lead to a reduction of specific secreted miRNAs in the extracellular environment [32]. Although circulating miRNAs are incapsulated in membrane-enclosed vesicles, the majority are bound to protein complexes. In the study of Arroyo et al., a series of miRNAs, including miR-16, miR-92a, and let-7a, were detected in AGO2 immunoprecipitates, supporting the function of AGO2 ribonucleoprotein complexes as a significant vehicle of circulating miRNAs [41]. Besides AGO2, another RNA-binding protein that drives miRNA export, and localized in the nucleolus, is Nucleophosmin (NPM1). It plays a key role in the ribosomal RNA processing and binds extracellular miRNAs, protecting them from RNase degradation [42,43].

Nevertheless, it is noteworthy to consider that miRNA profiles can show a different distribution between donor cells, microvesicles, and recipient cells. Indeed, in order to favor the tumor microenvironment, the parental cancer cells preferentially release or retain specific miRNAs, on the basis of their target genes. For example, in breast cancer, nearly 90% of miR-451 is released in the extracellular environment, showing an enrichment interstitially compared to mammary epithelia. Indeed, since tumor suppressor miR-451 downregulates macrophage migration inhibitory factor (MIF) and multi-drug resistance 1 (MDR1), the malignant parental cells selectively release it in body fluids [44]. These findings suggest that cells might retain or release specific miRNA molecules, supporting the tumoral environment. Considering all these findings, as well as miRNA profiling in tissues, circulating miRNA in the extracellular environment might be used as a diagnostic biomarker for the screening and detection of several tumors.

## 3. Deregulation of Circulating miRNAs in Cancer as Potential Tumor Biomarkers 

### 3.1. MiRNAs in Serum and Plasma Samples

Many studies have demonstrated the deregulation of circulating miRNAs into biofluids in different types of cancer. miRNA profiling analysis in saliva, urine, breast milk, colostrum, tears, and seminal fluid are obtained by non-invasive techniques compared to those in plasma, cerebrospinal fluid, pleural fluid, and amniotic fluid, which require more invasive methods [45]. Serum and plasma show a greater series of stable miRNAs derived from tissues and organs and they represent the most studied biofluid. The first discovery of circulating miRNA in serum was observed in patients with diffuse large B cells lymphoma (DLBCL), which showed a high content of three miRNAs: miR-21, miR-155, and miR-210. In particular, the overexpression of miR-21 has been associated with the improvement of relapse free-survival. However, another group did not confirm a significant correlation with this clinical outcome. This association between miR-21 and the better prognosis of several DLBCL cases might be due to its ability to target genes involved in cell proliferation and tumorigenesis, including STAT3, E2F3, JAG1, and SKI [46]. However, the circulating miR-21 level displayed a different expression in other types of cancers. The overexpression of miR-21 and miR-222 in the plasma samples of gastric cancer patients compared to healthy controls has been demonstrated [47]. The altered level of miR-21 in the serum and plasma of many tumors, such as gastric cancer, is associated with its oncogenic role; it regulates migration and metastasis processes by downregulation of several tumor suppressor targets, such as PTEN, PDCD4, and SATB1. Another study on gastric cancer by Chen et al. revealed the overexpression of circulating plasma miR-196a in patients with precancerous lesions/early gastric adenocarcinoma compared to healthy groups. They divided the patients with intestinal metaplasia, low/high-grade dysplasia, and early gastric cancer and observed a higher and similar expression of miR-196a in all stages of precancer compared to healthy individuals [48]. MiR-206 showed a lower level in the serum of gastric cancer patients, correlating with the clinicopathological features (deep local invasion and tumor-node metastasis) and the high rate of recurrence of the tumor [49]. Tumor-associated circulating miRNAs have also been observed in non-small-cell lung cancer (NSCLC), which represents one of the most aggressive cancers, with high mortality. The noninvasive detection of circulating miRNAs in early stages of lung cancer can avoid the progression of disease and reduce the mortality of patients. Recent studies have discovered different plasma and serum miRNA signatures in the early stage of NSCLC. For example, the study of Ying et al. identified a panel of five miRNAs in serum that showed a different expression between NSCLC and healthy individuals. The early stage of NSCLC patients is characterized by a lower level of miR-let7a and miR-375 and a higher level of miR-1-3p, miR-1291, and miR-214-3p than healthy controls [50]. Geng et al. detected five novel miRNA biomarkers in the plasma of subjects affected by NSCLS at early stages. From the analysis carried out on 126 NSCLC patients, 42 NCPD patients, and 60 healthy controls, some miRNAs (miR-20a, miR-145, miR-21, miR-223, and miR-221) were upregulated in NSCLS patients compared to two groups of controls [51]. An interesting study on plasma circulating miRNA in small cell lung cancer (SCLC) revealed the association of five miRNAs (miR-92b, miR-146a, miR-375, miR-1224, and miR-1246) with the response to the chemotherapy. These miRNAs were significantly higher in patients resistant to chemotherapy. Among these five miRNAs, a high level of miR-375 and miR-92b was associated with shorter disease free survival, supporting their role as biomarkers for the chemotherapy response and prognosis of SCLC patients [52]. Breast cancer is another tumor where nearly 30% of patients are characterized by recurrence, leading to the formation of local and distant metastasis. Papadaki et al. analyzed the miRNA plasma level in patients with disease free survival, early (within 3 years post-surgery) and late relapse (5 years post-surgery) during the follow up period. They found that miR-190 showed a lower expression in patients with early relapse, while miR-21 and miR-200c were higher in those with late relapse compared to healthy patients. Additionally, miR-23b also showed a higher expression in patients with relapse of the disease. Therefore, different levels of circulating miRNAs reflect the dormant state of breast cancer [53]. Interestingly, a different level of miRNAs in serum among patients with benign and malignant breast disease has been demonstrated. Circulating serum miR-21 and miR-20a showed a high expression in breast and benign cancer compared to control groups; only miR-214 had a significant difference between benign and malignant disease. Furthermore, the level of miR-214 decreased after resection [54], suggesting its possible function as a potential biomarker in malignant disease. The role of miRNAs as a potential diagnostic biomarker has been studied in thymic epithelial tumors (TETs), a rare neoplasm that arises from the anterior mediastinum. Expression of six selected miRNAs (miR-21-5p, miR-148a-3p, miR-141-3p, miR-34b-5p, miR-34c-5p, and miR-455-5p) was evaluated in plasma samples from TET patients and healthy controls before and after surgery. Among these miRNAs, only miR-21 and miR-148a-3p were significantly upregulated in the plasma of TET patients compared to healthy controls. On the contrary, their expression was reduced during the follow up, supporting the function of these miRNAs as non-invasive biomarkers in thymic epithelial tumors [55]. Of note, many studies have demonstrated the relevance of circulating miRNA profiling expression for the detection and progression of colorectal carcinoma (CRC). The study by Zekri et al. identified some serum miRNAs, including miR-17, miR-19a, miR-20a, and miR-223, upregulated in the CRC group compared to the healthy group. The altered expression of these miRNAs is associated with their oncogenic function in colorectal cancer; they regulate the expression of genes involved in cell cycle progression, epithelial to mesenchymal transition (EMT), and cell growth and pro-survival pathways [56]. Moreover, many circulating serum/plasma miRNAs showed an association with the progression of CRC. For example, serum miR-103 was upregulated in CRC patients with lymph node metastasis, distant metastasis, and advanced tumor stage compared to control groups. In addition, miR-103 expression correlated with shorter overall survival [57]. Other circulating miRNAs that could be used as a biomarkers of CRC patient’s prognosis are miR-141, miR-200b, and miR-200c. These miRNAs are related to distant metastasis of CRC. SH. Zhu et al. analyzed the expression of serum miRNA in colorectal cancer patients with, and without, liver metastases. They observed a significantly higher expression of miR-141, miR-200b, and miR-200c in liver metastasis in colorectal carcinoma compared to the control group, suggesting these miRNAs as new serum biomarkers of this disease [58]. Moreover, miR-200b with three other circulating miRNAs, i.e., miR-29, miR-31, and miR-203, showed a higher level in the plasma of CRC patients compared to healthy controls; miR-29 and miR-31 are associated with a high risk of recurrence of the disease, indicating these miRNAs as relevant biomarkers for the prediction and detection of recurrence of CRC [59]. Since circulating miRNAs show a different expression between cancer and normal samples, they can act as diagnostic and prognostic biomarkers, allowing the early identification and the treatment of tumors.

### 3.2. miRNAs in Other Biofluids

Although the majority of studies have focused on serum and plasma miRNA expression profiling, recent research has also detected the alteration of miRNAs in other body fluids.

For example, in nasopharyngeal carcinoma (NPC) a panel of differently expressed miRNAs in salivary samples were detected. The NPC patients showed a higher expression of miR-3679-3p, miR-574-5p, miR-205-5p, and miR-6131, and a lower expression of 30b-3p, miR-575, and miR-650 compared to the control, suggesting these salivary miRNAs as diagnostic biomarkers for this disease [60]. Gastric juice can also have an important diagnostic value as a source of biofluid in the detection of prognostic and diagnostic biomarkers for gastric cancer. Interestingly, although miR-21 and miR-106a are overexpressed in serum samples, they showed a lower expression in the gastric juice of patients with gastric cancer compared to patients with benign gastric disease. In this case, it is possible that cancer cells might retain oncogenic miRNAs, like miR-21 and miR-206a, in order to avoid their release into the extracellular environment and to preserve the tumorigenesis process [61]. Moreover, the study of circulating miRNAs in serum and in urine specimens has shown the same level of expression in patients with breast cancer compared to healthy controls, highlighting their potential role as useful biomarkers of disease. For example, miR-424, miR-125b, and miR-194 are upregulated both in urine and serum tumor samples. The combination of miR-424 and three other miRNAs (miR-423, miR-660, and let-7i) might be useful as a biomarker tool to distinguish breast cancer patients from healthy controls. Furthermore, the level of the miR-7 family (including miR-7a, miR-7d, and miR-7f) is considered an important diagnostic biomarker in breast cancer, given their similar expression in serum and urines samples [62]. As in breast cancer, the analysis of miRNA profiling expression in the urine samples of ovarian cancer patients has led to the identification of specific miRNAs as prognostic biomarkers. Zhou et al. found the upregulation of miR-30a-5p in the urine of patients with ovarian serous adenocarcinoma compared to healthy controls and its association with the early stage of tumors [63]. Moreover, seminal fluid represents a promising biofluid for the detection of prostate cancer (PCa). A recent study by Barcelò et al. identified the upregulation of three miRNAs (miR-142-3p, miR-142-5p and miR-223-3p), combined with the serum level of prostate-specific antigen (PSA), in PCa patients compared to healthy controls. Moreover, miR-342-3p in seminal fluid showed a different expression between G6 and G7 grades of PCa patients; emerging as a biomarker for the progression of prostate cancer [64]. As we have described above, miR-21 showed aberrant expression in the serum and plasma of several cancer samples, confirming its role as a prognostic and diagnostic biomarker of different pathologies. Nonetheless, the analysis of miRNA profiling in the cerebrospinal fluid of patients affected by glioma revealed the higher level of miR-21 in glioma groups than in healthy individuals. Importantly, the level of miR-21 correlates with the clinical features of tumors, demonstrating a more elevated expression in the high-grade glioma group compared to the low-grade glioma group. Furthermore, the level of miR-21 in cerebrospinal fluid decreased after the resection of tumors [65]. 

On the basis of these findings, miRNA profiling analysis in other biofluid constitutes a novel tool for the diagnosis, prognosis, and monitoring of tumor progression.

## 4. Correlation of miRNA Levels in Liquid Biopsies and in Matched Tissue Samples 

MiRNAs differentially expressed in the extracellular fluids of cancer patients compared to healthy individuals are also usually deregulated in tumor vs. normal matched tissues [66] (Table 1). 

For example, the expression of miR-196a in gastric cancer is higher in the tumor tissue compared to its normal counterpart, as well as in the plasma samples of cancer vs. healthy individuals. An analysis using Kaplan-Meier estimators showed a positive significant correlation between the overexpression of miR-196a and short survival of patients [67]. Moreover, as we described above, gastric cancer is characterized by the downregulation of circulating miR-206. The downregulation of miR-206 has been observed in gastric cancer tissues compared to the corresponding non-cancerous tissues, confirming the tumor suppressor role of miR-206 and its biomarker function [68]. Although miR-21 and miR-106a showed a lower expression in the gastric juice of tumor patients, they are similarly expressed in both cancer tissue and in the serum [70]. This evidence is consistent with the ability of tumor cells to retain specific oncogenic microRNAs, promoting progression of pathogenesis. MiR-21 is the most common oncogene microRNA upregulated in different types of cancer, such as lung [72], breast [88], gastric cancer [69], and glioma [71]. In these tumors, different studies have demonstrated the upregulation of miR-21 in cancer tissue compared to the normal counterparts. 

Interestingly, microRNA expression profiling studies in NSCLC tissues have shown a similar trend with circulating microRNA levels in plasma, such as miR-20a [73], miR-221 [75], and miR-223 [74]. On the contrary, miR-145 [76] is significantly downregulated in NSCLC tissue, showing a different expression with respect to its level in plasma specimens. A different expression of microRNAs between tissue samples and biofluids has been identified in TETs. Although miR-34b-5p, miR-34c-5p, and miR-455-5p are undetectable in the plasma of TET patients and healthy donors, they are upregulated in tumor tissues. Moreover, miR-34b-5p, miR-21-5p, miR-148a-3p, miR-141-3p, and miR-205-5p expression is higher in metastases compared to tumor and normal tissue [25]. In addition, in breast cancer, the circulating microRNAs, such as miR-214, showed an opposite expression in the tumor tissue. The study of Liu et al. reported the downregulation of miR-214 in breast cancer tissue compared to healthy counterparts, suggesting a tumor suppressor role of this miRNA [77]. Interestingly, miR-190 showed a low expression in breast cancer tissue and was correlated with the better survival of patients. The overexpression of miR-190 inhibits breast cancer metastasis both in vitro and in vivo [78]. These data are consistent with the lower expression of serum miR-190 in relapsed, compared to non-relapsed, patients, as we described above. On the contrary, miR-424 [79] and miR-125b [80] are upregulated in urine and serum in human breast cancer, but are downregulated in tumoral tissues, supporting their ability to impair cell proliferation and breast cancer progression. MiR-let-7a has a similar expression in biofluids and tissues. Tumor tissues collected from breast cancer patients at advanced stages are characterized by the downregulation of miR-let7a expression, confirming its role in tumor suppression [81]. 

In CRC, miR-19a and miR-20 showed an upregulation in CRC tissues compared to the respective healthy counterparts. Both microRNAs belong to the miR-17-92 cluster, which is one of the oncogenic microRNA family. In particular, high miR-19a expression correlates with lymph node metastasis and TNM stage [82]. The study of Zheng et al. reported the overexpression of miR-103 in CRC tissue and its association with a metastatic phenotype. The downregulation of miR-103 in CRC cell lines led to the reduction of cell proliferation, invasion, and migration [83]. As we reported, the miRNA-200 family (which contains miR-200a, miR-200b, miR-200c, miR-141, and miR-429) is involved in CRC and is used as a diagnostic and prognostic serum biomarker. The study of Diaz et al. observed an upregulation of all five members of the miR-200 family in tumors compared to normal tissue. Moreover, the expression of these miRNAs increased in parallel with the stage of CRC [84]. Paperson et al. reported a different expression of the miR-200 family in CRC tissue. Indeed, micro-dissected colorectal tissue showed a low miR-200 expression at the invasive front of tumors with a degraded basement membrane, characterized by EMT phenotype. Instead, regional lymph node metastases and vascular carcinoma reported a higher expression of miR-200, suggesting that these microRNAs play a relevant role in the formation of metastasis [85]. 

A correspondence of miRNA expression between biofluids and tissues was observed in prostate cancer for miR-142-3p and miR-223-3p. Both miRNAs are upregulated in PCa tissue compared to non-cancer tissue. Indeed, the overexpression of miR-142-3p and miR-223-3p in PCa cell lines increases cell proliferation and cell cycle progression targeting the FOXO1 and SEPT6 genes, respectively [86,87]. Finally, the study of Zhou et al. demonstrated that a high level of miR-30a-5p in the urinary of ovarian serous adenocarcinoma is correlated to its expression in tissues. Authors showed a significant overexpression of miR-30a-5p both in tissue samples collected from ovarian serous adenocarcinoma patients and in four ovarian cancer cell lines. Moreover, the reduction of miR expression after surgery supported its association with ovarian serous adenocarcinoma, confirming its origin from the ovarian cancer tissue [63].

Altogether, these data highlighted the importance of evaluating matched samples of liquid biopsy and tumor tissue as these biological samples might present concordant or discordant expressions of miRNAs, depending on the organ involved and type of sample analyzed. 

## 5. Deregulation of miRNAs in Renal Cell Carcinoma: A Focus on Clear Cell Renal Cell Carcinoma

Renal cell carcinoma (RCC) represents the third most common urologic cancer and approximately 90% of all kidney malignancies with high mortality rate. It arises from renal tubular epithelial cells, including morphologically and genetically heterogeneous subtypes: clear cell renal cell carcinoma (ccRCC, 70–80% of cases), papillary renal cell carcinoma (pRCC, 10–15% of cases), and chromophobe renal cell carcinoma (chRCC, 5–10% of cases). Since these tumors are asymptomatic, they are diagnosed at late stages of disease, often characterized by distant metastases. RCC treatment requires partial or radical nephrectomy, ablative therapies, and active surveillance for located RCC, whereas therapeutic treatments are used for metastatic RCC [89]. Given the great heterogeneity of RCC, many studies have analyzed miRNA expression profiling to distinguish different types of this tumor, suggesting that their altered expression could have a diagnostic and prognostic power. 

### 5.1. Deregulation of miRNAs in ccRCC Biofluids

Among the RCC, we have focused our attention on circulating miRNAs in serum, plasma, and urine samples of ccRCC, which is the most common subtype of RCC (Table 2). Of note, many circulating miRNAs have been detected as non-invasive biomarkers in both serum and plasma of ccRCC. One of the most important miRNAs is miR-210. Iwamoto et al. discovered the higher level of serum miR-210 in patients at the early stage of ccRCC compared to healthy controls [90], suggesting a prominent role of miR-210 as a biomarker for the early diagnosis of this disease. In agreement with these data, a recent study found the upregulation of miR-210 combined with two other miRNAs, miR-1233 [91] and miR-155 [92], in ccRCC patients compared to the control group. The downregulation of both miR-210 and miR-1233 after surgery, indicated both as good biomarkers of cancer prognosis. These studies did not demonstrate a correlation between the serum level of miR-210 and the clinicopathological characteristics. However, Nakada et al. reported an association between miR-210 and hypoxia environment: an upregulation of miR-210 occurred after accumulation of hypoxia-inducible factor 1α (HIF1α) [93]. Heinemann et al. performed an analysis of serum miRNA profiling in an independent cohort, including patients with ccRCC, benign renal tumors (including angiomyolipoma, oncocytoma, and cystis), and healthy individuals. They have discovered lower levels of two miRNAs, miR-206 and miR-122-5p, in ccRCC and benign renal tumors groups compared to controls. On the other hand, they have observed an increase of serum miR-206 and miR-122-5p level in patients with metastatic disease. Furthermore, they demonstrated a correlation between high expression of these miRNAs and shorter disease-free and overall survival, confirming their relevant role as potential prognostic biomarkers [94]. Teixera et al. demonstrated the association of miRNAs with poor overall survival of renal carcinoma. This study was based on a relative quantification of miR-221 in 77 plasma samples, and carrying out a follow-up of the patients. The overexpression of miR-221 and miR-222 in RCC patients with and without mestastasis and their significant correlation with lower overall survival, supported the hypothesis that they might help the diagnosis of RCC, as noninvasive biomarkers of tumor development [95]. Together with miR-210; miR-193a-3p, miR-362, and miR-572 were also significantly increased, whereas miR-28-5p and miR-378 were decreased in patients with RCC compared with the control groups [96]. Most of the circulating miRNAs were identified during the early stage of RCC. However, Chanudet et al. described different plasma miRNAs associated with the later stages of ccRCC. Analysis of miRNAs in 94 RCC cases and 100 healthy controls, revealed a significantly different expression of specific miRNAs between healthy controls and patients at early and late stage of ccRCC, suggesting their involvement in the progression of the disease. Specifically, miR-451 seem to be downregulated in the stage III and VI ccRCC patients compared to the control groups and earlier stage of ccRCC. The association of lower plasma levels of miR-451 in advanced stages of ccRCC was explained by its tumor suppressor activity, involved in the regulation of the PI3K/Akt/mTOR signaling pathway in different types of cancer. In agreement with the aberrant expression of miR-451, late stage ccRCC cases are characterized by the overexpression of target genes of downregulated circulating miRNAs in plasma. Conversely, the same downregulation of the miR-451 in ccRCC tissue is associated with a worse response to chemotherapeutic treatment [97], highlighting that the regulation of a defined miRNA is often different between plasma and tissue. These genes belong to oncogenic pathways such as PI3K/Akt, ErbB, and HIF-1, activated in renal tumors [98]. Both serum and plasma-derived miRNAs may have a prognostic value. For example, Du et al. investigated the expression of plasma circulating miRNAs in metastatic renal cell cancer (mRCC), representing a pioneer study that quantified the circulating plasma miRNAs (miR-190b, miR 26a-1-3p, miR-let-7i-5p, miR-145-3p, miR-200-3p and miR-9-5p, and miR-615-3p) associated with the overall survival of mRCC patients. Low levels of miR-let-7i-5p, miR-26a-1-3p, and miR-615-3p are associated with poor overall survival [99], and they are emerging as predictive candidates for the progression and development of cancer. 

Considering different body fluids for the detection of miRNAs biomarkers, renal cell carcinoma, and other urological tumors highlights the significance of urine samples as a novel promising source for the identification of potential new miRNAs biomarkers. Indeed, the analysis of urine has some important advantages: patient’s samples are obtained rapidly and easily by non-invasive procedures and can be repeatedly obtained [30]. As described above, miR-210 is one of the most important biomarkers overexpressed in the serum and plasma of a variety of tumors, including RCC. Notably, starting from our previously published works highlighting the upregulation of miR-210-3p in ccRCC tissues compared to matched normal counterparts, we evidenced that miR-210-3p was significantly upregulated in urine specimens collected from two independent cohorts of ccRCC patients at the time of surgery compared to healthy donors samples [100,104]. Moreover, miR-210-3p levels resulted significantly reduced in follow-up samples [105]. In agreement with these results, Li and colleagues provided evidence that the level of urinary cell-free miR-210 was significantly higher in patients with ccRCC and significantly decreased in the patients a week after surgery [106]. All these data highlight that the level of miR-210-3p in urine specimens represents a potential and novel tool for the diagnosis and evaluation of treatment response during ccRCC follow-up and the monitoring of progression-free survival. 

Besides miR-210-3p, recent studies have investigated the expression of different miRNAs in urine specimens for the progression of ccRCC. For example, Cocchetti et al. investigated the level of specific miRNAs being overexpressed in ccRCC by data bank analysis. Among these miRNAs, they observed a high and significant level of miR-122, miR-1271, and miR-15b in ccRCC urine specimens compared to control groups. The high level of miR-122 in urine targets FOXP1 gene, correlating with oncogenic activity in renal carcinoma and favoring cell growth and tumorigenesis progression [101]. 

So far, we have discussed the aberrant expression of oncogenic miRNAs in urine specimens, but several studies have also identified differentially expressed tumor suppressor miRNAs between ccRCC patients and non-cancer subjects. The work of Song et al. showed a lower level of miR-30c-5p in ccRCC patients compared to control groups. Indeed, miR-30c-5p acts as a tumor suppressor in cancer, inhibiting the expression of the HSAP5 gene, which positively correlates with cancer growth, aggressiveness, and metastasis formation. On the basis of these data, the low expression of urinary exosomal miR-30c-5p is useful for the detection of early stage ccRCC [102]. An interesting mechanism that can promote the downregulation of tumor suppressor miRNA in cancer is the methylation of its promoter [107]. This molecular process might become a new tool for the detection of ccRCC. As demonstrated by Pinho et al., the level of promoter methylation of miR-30a-5p is elevated in ccRCC and metastatic urine samples compared to healthy donors and non-metastatic ccRCC, suggesting that miR-30a-5p^me^ level might be an indicator of disease progression and metastatic processes [103]. Finally, the second most common type of RCC is papillary renal cell carcinoma (pRCC), which represents 10–20% of all renal cell cancers. Unlike the studies in ccRCC, the analysis of circulating miRNA as diagnostic biomarkers in pRCC has been restricted to a few patients; as a result, little information is available concerning miRNA profiling expression in body fluids. Kalogirou et al. investigated serum miRNA expression in a cohort including 67 pRCC patients and 33 healthy individuals. Based on The Cancer Genome Atlas (TCGA) pRCC dataset, they selected 11 miRNAs for serum analysis and found a significantly higher level of miR-21-5p in advanced pRCC compared to healthy donors, whereas miR-210-3p showed a downregulation in localized pRCC. Both serum miRNAs could be clinically relevant as diagnostic and prognostic markers in pRCC [108]. 

### 5.2. Deregulation of miRNAs in ccRCC Tissue

To date, the expression of a wide array of miRNAs in neoplastic tissues is well acknowledged in almost all types of cancer, including ccRCC. From a biological point of view, cancer is extremely heterogenous, and the biological changes occurring over time allow the tissue to acquire significant characteristics and properties, including a parallel alteration in gene expression and miRNAs landscape. 

Thus, several miRNAs are emerging as relevant players involved in the molecular circuitry at the basis of ccRCC phenotype. According to different bioinformatics platforms, experimental settings, in vitro or in vivo correlations, and clinical parameters, different studies have revealed distinctive miRNAs signature associated with ccRCC. By employing diverse experimental systems and datasets, some studies have shown that 63 miRNAs were altered in 516 patients with ccRCC [109], whereas others have reported between 11 and 54 miRNAs [110] when cancer tissue is compared to normal samples [111]. Nevertheless, the literature, at least, clearly highlights that miRNAs can foster or inhibit cancer properties according to their ability to target oncogenes, as well as oncosuppressors (*EGFR*, *mTOR*, *VHL*, *HIF-1α*, *PDGFβ*, *TP53*, *PTEN*, *BAP1*, *SETD2*, *PBRM1*, *KDM5C*) [112]. For instance, miR-216b can suppress the expression of *KRAS*, by binding the 3′ UTR of the gene. This results in the inhibition of KRAS, which is frequently mutated and associated with proliferative pathways, such as MAP/ERK and PI3K/AKT [113]. Similarly, miR-223-3p, highly expressed in cancer renal tissue and positively correlated with ccRCC staging and progression, downregulates at protein level the solute carrier family 4, member 4 (SLC4A4), which also regulates KRAS, therefore favoring proliferation and metastasis [114]. 

Differentially expressed analysis of both genes and miRNAs in ccRCC has highlighted that more than 4000 genes are upregulated or downregulated in tumor tissue compared to non-tumoral adjacent cancer tissue and that a small group of miRNAs including miR-1, 133a, 144 and 21 are commonly expressed by other types of tumors, strengthening the role of miRNAs as homogenous biomarkers [115]. Despite this, the study also highlighted that the negative correlation of critical miRNAs and target genes, linked to carcinogenesis and progression, was decreased in ccRCC, as well as in the other types of tumors analyzed. Thus, the interaction of miRNAs and gene targets could be influenced by environmental or exogenous factors, altering both their expression and, more importantly, their biological function. 

The majority of studies almost unanimously describe and confirm that in ccRCC a wide landscape of miRNAs is mainly downregulated. The in vitro knockdown of these miRNAs implies the loss of inhibitory targets, which are at the crossroad or downstream molecules of several biological pathways, ranging from basic biological functions, such as cell proliferation and migration, to modulation of main metabolic process and inflammation. In other words, these miRNAs would exert tumor-suppressive effects in ccRCC.

Nevertheless, starting from the differential expression of miRNAs in ccRCC tissue, the majority of studies have validated in vitro a long list of miRNAs only by employing stabilized renal cell lines or by bioinformatics prediction, algorithms, atlases, and often heterogenous clinical/histological criteria among reports, therefore hampering the harmonization of findings with their clinical significance and the translatability of miRNAs involved in ccRCC. Although in a marginal group, an additional set of miRNAs can be found upregulated in parallel, and functioning as tumor-enhancers rather than tumor suppressors as in their downregulated counterparts. 

In the light of this, the miRNAs unequivocally responsible for the pathogenesis and progression of ccRCC have not been clearly identified. A possible reason is due to the heterogeneity and staging of ccRCC itself, also accompanied by the clinical features of patients, including immune surveillance and more importantly the genetic background and mutational assets. Consequently, the overall, huge amount of data is still inconsistent, and we are far from a complete clinical applicability and therapeutically based miRNA strategy in ccRCC. Besides, as miRNAs always act as a more complex network, it is still unclear how the whole panel of downregulated and upregulated miRNAs might orchestrate the etiopathogenesis and progression of ccRCC.

Despite this, specific miRNAs are more frequently found to be dysregulated in ccRCC, and molecular mechanisms and axes have been identified, although a huge variability exists, and the experimental approaches are mainly performed trough bioinformatics algorithms, atlases, and often heterogenous clinical/histological criteria among studies. For instance, miR-141 and miR-200c are certainly the most deregulated miRNAs in ccRCC, a mechanism likely linked to the inhibition of the CDH1/E cadherin pattern involving the enhancement of the transcriptional factor ZFHX1B [116].

More importantly, a single miRNA can target multiple and distinctive pathways. For instance, the overexpression of miR-21 in ccRCC is associated with hypomethylation at the CpG locus in its promoter, which also represents a target site of CEBPB, MEIS3, and TEAD4, the main transcriptional factors involved in embryonic development and EMT during tumor progression [117,118,119]. However, the upregulation of the same miR-21 and the resulting renal dysfunction in ccRCC could also be ascribable to the activation of a different molecular axis, involving the uremic toxin p-Cresyl sulfate, which fosters a hypoxic microenvironment trough HIF-1α [120]. Both miR-645 and miR-654 support ccRCC cell proliferation and progression, and have also been found upregulated in ccRCC tissues. Interestingly, they both act through the glycerol kinase-5 (GK5) [121]. miR-122 directly targets FOXO3 (a main transcriptional factor involved in ccRCC metastasis [122], behaving as a tumor-promoting miRNA [123]. 

Profound epigenetic modifications, mainly linked to the level of methylation, seem important for determining the downregulation of defined miRNAs involved in ccRCC. Advanced stages of the tumor are negatively correlated to the mRNA levels of Methyltransferase-like 14 (METTL-14), sequestered by circulating RNAs, and resulting in the methylation of *PTEN*, a frequently mutated gene in ccRCC [124,125].

Recently, the Cullin 4B protein, a scaffold protein involved in the cullin-RING ubiquitin ligase (E3) complexes and tumorigenesis [126], was found to be upregulated in ccRCC, influencing apoptotic and cell survival signaling. Cullin 4B is a target of miR-217, which is downregulated in ccRCC and negatively associated with Cullin 4B, by inhibiting its expression [127]. MiR-483-5p is downregulated in ccRCC compared to normal tissue. Lower stages of ccRCC express increased levels of miR-483-5p, also correlating with decreased neutrophils/lymphocytes and monocyte/lymphocytes ratios, demonstrating a negative association with the immune response, inflammation, EMT, and metastatic progression [128]. Different stages of ccRCC have been proved to correlate with different expression profiles of miRNAs. In pT1 ccRCC, miR-15a-5p and -26a-5p have been found to be altered. When overexpressed in vitro, both miRNAs decrease the expression of the O-GlcNAc-transferase, an enzyme catalyzing the addition of O-GlcNAc to proteins as a post-translational modification with similar effects to phosphorylation [129] and therefore considered an oncogenic factor in ccRCC [130]. The upregulation of miR-153-5p in ccRCC is associated with a poor clinical prognosis. The mechanism involves the targeting of AGO1, a key protein physiologically participating in mechanisms of RNA silencing, including miRNAs [131] and a predictive marker of ccRCC [132]. The miR-153-5p/AGO1 axis has been demonstrated to act through the PI3K/AKT pathway, and therefore fostering tumor growth and progression [133].

In ccRCC the role of miRNAs is also related to physiological metabolic process, which are normally modified in cancer, to allow growth independence and resistance to harsh microenvironments. Accordingly, the loss of von Hippel-Lindau protein (pVHL) and hypoxia inducible factor (HIF) activation have been implicated in the Warburg effect observed in several tumors and in particular as a hallmark of ccRCC, where aerobic glycolysis is dysfunctional, as well as the accumulation of lipids as a consequence of hypoxia [134,135,136]. Accordingly, a well acknowledged panel of different miRNAs are known to participate in both glycolytic and fatty acids biochemical metabolisms: miR-22, 340, 1291, 144, 132, 186, and 495 negatively regulate the GLUT-1 protein; whereas Let-7, miR-106b, 494, 93, 223, 133, 199a, 146a, 31, 29a, 29c, 27a, and 30d are involved in GLUT-4 modulation [134]. The list of miRNAs for the glycolysis pathways is even more numerous, also including miR-122 and 16 for the aldolase and miR-146 and 199 for control of hexokinase 2. The miR-765 has been observed to be increased in the plasma of patients with ccRCC after surgery, likely acting as a suppressor of the proteolipid protein 2 (PLP2) and of the accumulation of lipids in the renal tissue [137].

Recently, the contribution of miRNAs to drug resistance or immunotherapy in ccRCC has been also investigated. Interestingly, sunitinib, the small-molecule multi-targeted receptor tyrosine kinase (RTK) inhibitor, and the most employed agent for treating ccRCC [138], induces an in vitro upregulation of miR-15b, leading to both decreased levels of cell apoptosis, Cyclin C, and cyclin dependent kinases levels (CDK19 and CDK8) [139]. If resistance to mTOR inhibitors occurs in everolimus-resistant cell lines, intracellular levels of miR-101 are downregulated with an increased excretion in the extracellular milieu [140]. The downregulation of miR-381 in renal tissues, but also in Caki-1 and 786-O metastatic renal cell lines [141], sustains in vitro cell proliferation, migration, and chemoresistance to paclitaxel, cisplatin [142], or to 5-fluorouracil targeting the cyclin-dependent kinase 2 [143]. Similarly, the downregulation of miR-497-5p has been correlated to the upregulation of PDL-1, which is overexpressed in ccRCC and is a well acknowledged target for immunotherapy. The miR-497-5p is able to decrease the in vitro clonogenic capacity of ccRCC and foster apoptosis in renal cell carcinoma [144]. 

Alternative molecular mechanisms can indirectly involve miRNAs by different methods, such as the transferring of miRNAs through the exosomal cargo, the circular, or even the long non-coding, RNAs. All these alternative mechanisms are reported to target the physiological processes responsible for the etiopathogenesis of the ccRCC, such as hypoxia, EMT phenomena, chemoresistance, or metastasis (mainly long non-coding RNA). For instance, the miR-19b-3p released in ccRCC by exosomes in the microenvironment is responsible for both activating EMT via r PTEN and cross talking with long distant niches to favor metastasis [145]. The circular RNA, named circPVT1, is upregulated in ccRCC tissue according to the histopathology [146]. CircPVT1 directly binds the miR-145-5p. This complex interferes with the transcriptional factor TBX15, the same downstream target of a different miRNA, miR-212-5p, which in turn behaves as a tumor suppressor gene in ccRCC [147]. A similar mechanism, involving the binding of miR-296-3p and linking to the regulation of the E-cadherin, is the circ-AKT3, whose downregulation has been associated with ccRCC metastasis [148]. In addition, the androgen receptor (main role in the vasculogenesis of ccRCC [149]) has been reported to increase the expression of circRNA HIAT1, involving the modulation of a more complex molecular axis, where multiple miRNAs (miR-195,-5p,/29a-3p/29c-3p) act on the Cell division control protein 42 (CDC42), leading to metastases inhibition [150]. Is it well acknowledged that patients with ccRCC are more prone to relapse and metastasis. Similarly to the effect of circ HIAT1 on the cell cycle of ccRCC, the suppressive action of the miR-4429 has been associated with the negative regulation of the cyclin-dependent kinase 6 (CDK6) and to decrease fibrosis [151].

Other reports highlight that the long non-coding PCED1B-AS1 inhibits the miR-484, and therefore promoting ZEB1 expression [152], whose importance has been elucidated as the main transcription factor implicated in the EMT, metastasis, and chemoresistance in several tumors [153]. Similarly, ZEB2 is synergistically and negatively coregulated with miR-124 and miR-203, contributing to attenuating the EMT process [154]. The fibrosis developed by the deregulation of the epithelium and stroma components, represents a significant pathogenetic aspect of ccRCC that is highly associated with organ failure and patient survival [155]. 

The HOXA Transcript Antisense RNA Myeloid-specific 1 (HOTAIRM1), a long non-coding RNA critical for kidney differentiation, and the ADAMTS9-AS2 are both downregulated in ccRCC by hypoxia in the form of the spliced cytoplasmic HM1-3 isoform [156] and via miR-27-a3p-mediated modulation of FOXO1, respectively [157]. 

Moreover, the biogenesis of miRNAs themselves has been considered as equally important. Surgical samples of patients with ccRCC, as well as metastatic and non-metastatic renal cell lines, express low levels of Dicer (key ribonuclease responsible for microRNA biogenesis [158]) according to the degree of severity [159]. Similarly to the in vitro knockdown of miRNAs, the silencing of Dicer mimics oncogenic biological effects such as cell growth, invasion, and metastatic features. 

## 6. Conclusions

Circulating miRNAs in liquid biopsy, in parallel to the miRNAs deregulated in pathological tissues, are emerging as relevant clinical biomarkers for the management of several tumors, included ccRCC. The molecular stability of circulating miRNAs in liquid biopsy as plasma, serum, and urine has great potential, which renders them good tools to be novel, important, and non-invasive biomarkers for the detection and progression of disease. Overall studies have shown that the deregulation of defined miRNAs, both in tissue and in fluids, has a biological significance in ccRCC. Despite this, a certain discrepancy between circulating miRNAs and those expressed in the renal carcinoma tissue is observed, but also expected. It is likely that different miRNAs will be differentially expressed according to the different phases and staging of the disease; changing the landscape of their regulation over the time. In the future, this will imply the development of more distinct panels of validated miRNAs, similarly to the molecular panels, which have been designed to detect the genomic alterations in tumors for targeted therapies-based clinical purposes. Considering that the etiopathogenesis of ccRCC is mainly related to hypoxia, the miRNAs modulating this process and targeting HIF-1α, or in presence of key genomic alterations of this type of tumor (*VHL*, *c-MET*, *PTEN*, *p53*, *Succinate Dehydrogenase/Fumarate Hydratase*), will be preferable. The dual information derived from the genomic mutational assets and the panel of miRNAs, will provide both potential targets for identification, but it will be also useful to understand and monitor the modality by which ccRCC progresses over time in patients. 

## Figures and Tables

**Table 1 genes-12-01154-t001:** Correlation of miRNA levels in liquid biopsies and in matched tissue samples.

miRNA	Tumor Type	Expression in Tissue	Expression in Biofluids	References
miR-196	Gastric cancer	Upregulated	Upregulated in plasma	[48,67]
miR-206	Gastric cancer	Downregulated	Downregulated in serum	[49,68]
miR-21	Gastric cancer	Upregulated	Downregulated in gastric juice	[61,69]
miR-106	Gastric cancer	Upregulated	Downregulated in gastric juice	[61,70]
miR-21	Glioma	Upregulated	Upregulated in cerebrospinal fluid	[65,71]
miR-21	Non-small cell lung cancer	Upregulated	Upregulated in plasma	[51,72]
miR-20a miR-223-3p miR-221-3p	Non-small cell lung cancer	Upregulated	Upregulated in plasma	[51,73,74,75]
miR-145	Non-small cell lung cancer	Downregulated	Upregulated in plasma	[51,76]
miR-34b-5p, miR-34c-5p, miR-455-5p miR-141-3p, miR-205-5p miR-21-5p, miR-148a-3p	Thymic Epithelial Tumors	Upregulated Upregulated	Undetectable in plasma Upregulated in plasma	[25,55]
miR-214	Breast cancer	Downregulated	Upregulated in serum	[54,77]
miR-190	Breast cancer	Downregulated	Downregulated in serum	[53,78]
miR-424 miR-125b	Breast cancer	Downregulated	Upregulated in urine and serum	[62,79,80]
miR-let7a	Breast cancer	Downregulated	Downregulated in urine and serum	[62,81]
miR-19a miR-20	Colorectal carcinoma	Upregulated	Upregulated in serum	[56,82]
miR-103	Colorectal carcinoma	Upregulated	Upregulated in serum	[57,83]
microRNA-200 family (miR-200a, miR-200b, miR-200c, miR-141 and miR-429)	Colorectal carcinoma	Upregulated Downregulated in the invasive front of tumor with degraded basement membrane	Upregulated in serum	[58,84,85]
miR-142-3p and miR-223-3p	Prostate cancer	Upregulated	Upregulated in seminal fluid	[64,86,87]
miR-30a-5p	Ovarian cancer	Upregulated	Upregulated in urine	[52,63]

**Table 2 genes-12-01154-t002:** microRNAs in blood and urine as potential biomarkers in ccRCC.

miRNA	Sample	Significant Expression of miRNA	References
miR-210	Serum	Overexpression of serum miR-210 levels in RCC patients compared to healthy donors. The AUC was 0.77 (95% confidence interval, 0.65–0.89) and the sensitivity and specificity was 65% and 83%, respectively.	[90]
miR-210 miR-1233	Serum	Higher level of serum miR-210 and miR-1233 in ccRCC patients compared to controls. miR-210: the AUC was 0.69 (95% confidence interval, 0.61–0.77) and the sensitivity and specificity was 70% and 62,2%, respectively. miR-1233: the AUC was 0.82 (95% confidence interval, 0.75–0.89) and the sensitivity and specificity was 81% and 76%, respectively.	[91]
miR-210 miR-155	Serum	Higher level of serum miR-210 and miR-155 levels in patients with ccRCC than in healthy controls (HCs). Furthermore, only exosomal miR-210 showed a significant upregulation in patients with ccRCC vs. HCs. miR-210: the AUC was 0.87 (95% confidence interval, 0.79–0.95) and the sensitivity and specificity was 82.5% and 80%, respectively.	[92]
miR-122-5p miR-206	Serum	High level of miR-122-5p and miR-206, respectively, in metastasized ccRCC and in advanced pT-stage ccRCC. miR-206: the AUC was 0.73 (95% confidence interval, 0.616–0.849) and the sensitivity and specificity was 57.1% and 83.8%, respectively.	[94]
miR-221 miR-222	Plasma	High level of circulating miR-221 and miR-222 compared to healthy individuals. High level of miR-221 in patients with metastasis compared to individuals without metastasis. Correlation between high expression of miR-221 and low overall survival. miR-221: the AUC was 0.696 (95% confidence interval, 0.499-0.893) and the sensitivity and specificity was 72.5% and 33.3%, respectively.	[95]
miR-193a-3p miR-362 miR-572 miR-28-5p miR-378	Serum	This 5-miRNA panel showed a high level in the stage I and stage IV groups compared with the noncancer controls. The AUCs for the combination of the 5 miRNAs were 0.801 (95% CI, 0.731–0.871) and 0.797 (95% CI, 0.732–0.863) for stage I and stage I–II cases, respectively, with high sensitivity and specificity. Association of these miRNAs with overall survival.	[96]
miR-190b miR 26a-1-3p miR-let-7i-5p miR-615-3p miR-200-3p miR-9-5p	Plasma	Kaplan–Meier analysis confirmed the significant association of miR-let-7i-5p (95% CI, 0.21-0.84), miR-26a-1-3p (95% CI, 0.10-0.84), and miR -615-3p (95% CI = 0.11-0.54) with OS. The association of miR-9-5p and miR-190 with OS was not statistically significant.	[98]
miR-210-3p miR-21-5p miR185-5p miR-221-3p miR-145-5p	Urine	Upregulation of miR-21-5p, miR-210-3p, and miR-221-3p in ccRCC fresh frozen tissues compared to matched normal counterparts, while miR-185-5p and miR-145-5p did not show modulation. Upregulation of miR-210-3p in ccRCC urine compared to healthy controls. Reduction of miR-210-3p in the ccRCC samples after surgery in the follow up study.	[100]
miR-122 miR-1271 miR-15b	Urine	High and significant level of miR-122, miR-1271, and miR-15b in the ccRCC urine specimens compared to control groups. The AUC for the combination of the 3 miRNAs was 0.96 (95% confidence interval, 0.88-1.04) and the sensitivity and specificity were 100% and 86%, respectively.	[101]
miR-30c-5p	Urine	Low expression of miR-30c-5p in ccRCC patients compared to controls. The AUC was 0.8192 (95% confidence interval, 0.7388-0.8996) and the sensitivity and specificity was 68.57% and 100%, respectively.	[102]
miR-30a-5p^me^	Urine	Significantly higher miR-30a-5p^me^ levels in urine from ccRCC patients compared to asymptomatic controls. The AUC was 0.6837 (95% confidence interval, 0.5837-0.7837) and the sensitivity and specificity was 83% and 53%, respectively. Different expression of miR-30a-5p^me^ levels in urine between metastatic and non-metastatic ccRCC. The AUC was 0.7884 (95% confidence interval, 0.5505-0.8601) and the sensitivity and specificity was 80% and 71%, respectively.	[103]

Abbreviation: AUC (area under the curve).

## Data Availability

No new data were created or analyzed in this study. Data sharing is not applicable.

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
