# Peer review of "Circulating microRNAs from the Molecular Mechanisms to Clinical Biomarkers: A Focus on the Clear Cell Renal Cell Carcinoma"

_genes, 2021, doi:10.3390/genes12081154_

Round 1

Reviewer 1 Report

The review manuscript “Circulating microRNAs from the molecular mechanisms to clinical biomarkers: a focus on the clear Renal Cell Carcinoma” by Tito et al., summaries numerous recent findings of dynamic circulating miRNA effects on tumor development and progression. This review could provide a catalog of circulating miRNAs as a biomarker and contribute to clinical applications against certain tumors. However, there are many inconsistent citations of references, typos, inappropriate italics characters, misaligned tables as listed below. Authors should carefully revise the manuscript and correct them clearly to avoid misleading.   

Page1 line 34; RNAs transfer (tRNAs) will be transfer RNAs (tRNAs)

Page2 line 55; remove comma next to the “lead”  - - target mRNA lead - - -  

Page2 line 80; make a single period “biomarkers.”  

Page2 line 96; remove the first name “Rabinowits et al.,”

Page3 line 99; remove H. and add comma “ Valadi et al.,” 

Page3 line 107; add period “Hunter et al.,”

Page3 line 126; add period “Arroyo et al.,” 

Page4 line 165; add period “Chen et al.,” 

Page4 line 177; add period “Ying et al.,” 

Page4 line 180; add period “Geng et al.,” 

Page4 line 192; add comma “Papadaki et al.,”

Page4 line 196; make a single space between “healthy patients.”

Page5 line 214; add period “Zekri et al.,”

Page5 line 214-215; remove italic “miR-17, miR-19a, miR-20a and miR-223 unregulated in”

Page6 line 256; change to “let-7i”

Page6 line 261; add period “Zhou et al.,” 

Page6 line 265; add period “Barcelo et al.,” 

Table 1; 

remove all the first name initials in the references column to make it consistent with Table 2. 

Authors should make realignment of each row and adjust space to make it easy to read. 

Add descriptions for single and double asterisk and cross.  if there is no particular reason, remove the marks in the column of Expression in tissue and Expression in biofluids.    

Page8 line 301; make a single space between “as miR-20a”

Page9 line 310; remove the first name “Liu et al.,”

Page9 line 330; add period “Diaz et al.,”

Page9 line 332; make a single space between “normal tissue”

Page9 line 343; add comma “Zhou et al.,”

Table 2
Same as table 1 authors should make realignment each row and adjust space to make it easy to read. 

Page11 line 393; add comma “Teixera et al.,”

Page12 line 402; add comma “Chanudet et al.,”

Page12 line 417; remove the first name “Du et al.,”

Page12 line 425; make a single space between “of urine”

Page12 line 438; double-check reference it should be “[98]”

Page12 line 447; double-check reference 

Page13 line 455; double-check reference it should be “[100]”

Page13 line 458; remove the first name and add comma “Outeiro-Pinho et al.,”

Page13 line 465; make a single space between “fluids. Kalogirou”

Page13 line 466; add comma “Kalogirou et al.,”

Page13 line 468; make a single space between “for serum”

Page14 line 555; change to “miR-217”

Page15 line 602; change to “through”

Page16 line 647; change to “Despite”

Page25 line 1005-1006; remove the grant information from the reference “11 This work was - - - 81100483)"

Lastly, authors used both tumor or tumour in the text, use one of them. 

Author Response

Dear Prof. Alesse Guest Editor (special issue “The Role of MicroRNA in Cancer”),

thanks for all useful comments raised by Reviewers.

We are able now to submit a revised version of our manuscript (ID: genes-1288870).

Please, find below a point-by-point response.

In order to speed up the revision process, we have highlighted our corrections in yellow on the manuscript.

We hope in a favourable outcome.

Sincerely,

Prof. Antonella Calogero on behalf of all Authors

Sapienza University of Rome

Faculty of Pharmacy and Medicine

Department of Medical Surgical Sciences and Biotechnologies

C.so della Repubblica 79 04100 Latina

antonella.calogero@uniroma1.it

Tel. +39-07731757234

REVIEWER 1

Comments and Suggestions for Authors

The review manuscript “Circulating microRNAs from the molecular mechanisms to clinical biomarkers: a focus on the clear Renal Cell Carcinoma” by Tito et al., summaries numerous recent findings of dynamic circulating miRNA effects on tumor development and progression. This review could provide a catalog of circulating miRNAs as a biomarker and contribute to clinical applications against certain tumors. However, there are many inconsistent citations of references, typos, inappropriate italics characters, misaligned tables as listed below. Authors should carefully revise the manuscript and correct them clearly to avoid misleading.   

 COMMENT 1: Page1 line 34; RNAs transfer (tRNAs) will be transfer RNAs (tRNAs)

RESPONSE 1: the text has been edited, accordingly.

COMMENT 2: Page2 line 55; remove comma next to the “lead”  - - target mRNA lead - - -

RESPONSE 2: the text has been edited, accordingly.

COMMENT 3: Page2 line 80; make a single period “biomarkers.”  

RESPONSE 3: the text has been edited, accordingly.

COMMENT 4: Page2 line 96; remove the first name “Rabinowits et al.,”

RESPONSE 4: the text has been edited, accordingly.

COMMENT 5: Page3 line 99; remove H. and add comma “ Valadi et al.,” 

RESPONSE 5: the text has been edited, accordingly.

COMMENT 6: Page3 line 107; add period “Hunter et al.,”

RESPONSE 6: the text has been edited, accordingly.

COMMENT 7: Page3 line 126; add period “Arroyo et al.,” 

RESPONSE 7: the text has been edited, accordingly.

 COMMENT 8: Page4 line 165; add period “Chen et al.,” 

RESPONSE 8: the text has been edited, accordingly.

COMMENT 9: Page4 line 177; add period “Ying et al.,” 

RESPONSE 9: the text has been edited, accordingly.

COMMENT 10: Page4 line 180; add period “Geng et al.,” 

RESPONSE 10: the text has been edited, accordingly.

COMMENT 11: Page4 line 192; add comma “Papadaki et al.,”

RESPONSE 11: the text has been edited, accordingly.

COMMENT 12: Page4 line 196; make a single space between “healthy patients.”

RESPONSE 12: the text has been edited, accordingly.

 COMMENT 13: Page5 line 214; add period “Zekri et al.,”

RESPONSE 13: the text has been edited, accordingly.

COMMENT 14: Page5 line 214-215; remove italic “miR-17, miR-19a, miR-20a and miR-223 unregulated in”

RESPONSE 14: the text has been edited, accordingly.

COMMENT 15: Page6 line 256; change to “let-7i”

RESPONSE 15: the text has been edited, accordingly.

COMMENT 16: Page6 line 261; add period “Zhou et al.,” 

RESPONSE 16: the text has been edited, accordingly.

COMMENT 17: Page6 line 265; add period “Barcelo et al.,” 

RESPONSE 17: the text has been edited, accordingly.

COMMENT 18: Table 1; remove all the first name initials in the references column to make it consistent with Table 2.  Authors should make realignment of each row and adjust space to make it easy to read. Add descriptions for single and double asterisk and cross.  if there is no particular reason, remove the marks in the column of Expression in tissue and Expression in biofluids.    

RESPONSE 18: we have improved the alignment of both Tables. Asterisks and symbols have been removed. 

COMMENT 19: Page8 line 301; make a single space between “as miR-20a”

RESPONSE 19: the text has been edited, accordingly.

 COMMENT 20: Page9 line 310; remove the first name “Liu et al.,”

RESPONSE 20: the text has been edited, accordingly.

COMMENT 21: Page9 line 330; add period “Diaz et al.,”

RESPONSE 21: the text has been edited, accordingly.

COMMENT 22: Page9 line 332; make a single space between “normal tissue”

RESPONSE 22: the text has been edited, accordingly.

COMMENT 23: Page9 line 343; add comma “Zhou et al.,”

RESPONSE 23: the text has been edited, accordingly.

COMMENT 24: Table 2 Same as table 1 authors should make realignment each row and adjust space to make it easy to read. 

RESPONSE 24: we have realignment the Table

 COMMENT 25: Page11 line 393; add comma “Teixera et al.,”

RESPONSE 25: the text has been edited, accordingly.

 COMMENT 26: Page12 line 402; add comma “Chanudet et al.,”

RESPONSE 26: the text has been edited, accordingly.

COMMENT 27: Page12 line 417; remove the first name “Du et al.,”

RESPONSE 27: the text has been edited, accordingly.

COMMENT 28: Page12 line 425; make a single space between “of urine”

RESPONSE 28: the text has been edited, accordingly.

COMMENT 29: Page12 line 438; double-check reference it should be “[98]”

RESPONSE 29: the text has been edited, accordingly.

COMMENT 30: Page12 line 447; double-check reference 

RESPONSE 30: the text has been edited, accordingly.

COMMENT 31: Page13 line 455; double-check reference it should be “[100]”

RESPONSE 31: References have been checked

COMMENT 32: Page13 line 458; remove the first name and add comma “Outeiro-Pinho et al.,”

RESPONSE 32: the text has been edited, accordingly.

COMMENT 33: Page13 line 465; make a single space between “fluids. Kalogirou”

RESPONSE 33: the text has been edited, accordingly.

COMMENT 34: Page13 line 466; add comma “Kalogirou et al.,”

RESPONSE 34: the text has been edited, accordingly.

COMMENT 35: Page13 line 468; make a single space between “for serum”

RESPONSE 35: the text has been edited, accordingly.

 COMMENT 36: Page14 line 555; change to “miR-217”

RESPONSE 36: the text has been edited, accordingly.

COMMENT 37: Page15 line 602; change to “through”

RESPONSE 37: the text has been edited, accordingly.

 COMMENT 38: Page16 line 647; change to “Despite”

RESPONSE 38: the text has been edited, accordingly.

COMMENT 39: Page25 line 1005-1006; remove the grant information from the reference “11 This work was - - - 81100483)"

RESPONSE 39: the text has been edited, accordingly.

 COMMENT 40: Lastly, authors used both tumor or tumour in the text, use one of them. 

RESPONSE 40: we have used the word tumour consistently in the manuscript. 

Reviewer 2 Report

The authors provide a good and comprehensive overview on the literature on circulating miRNAs and their connection to cancer and specifically RCC. There is not much to add to this part, but the introduction, and especially the part on miRNA biogenesis has to be re-written. Please find specific comments below.

Although most of the text is written in good and clear English, the manuscript would somehow benefit by minor language editing possibly by an English native speaker. Some phrasing/word order does not sound right. Although, the reviewer has to admit not to be an English native speaker, so besides some corrections suggested below, probably has not found everything language/grammar-related.

General notes/comments/corrections:

„non-coding” is found with or without the “minus” several times in the text

The authors sometimes use capital and sometimes small “M” when starting a sentence with “miRNA”. Both is fine, but please be consistent.

There are several double spaces in the text, please use full text search to remove.

Specific corrections:

Line (“L”) 31: delete “they”

L39: the reviewer is unsure if miRNAs can be termed “epigenetic”

L42: better “approximately 170 nt”

L46: asterisk should be explained

L47: “most stable single strand“: too imprecisely written, please refer here for a better phrasing: https://onlinelibrary.wiley.com/doi/10.1002/wrna.1627

L47: The Ago protein has to be mentioned already in this section!

L53: to the reviewers knowledge miRNA binding to the 5’-UTR is rare, and some do bind to coding sequence. Please rephrase and use more recent literature to cite here!

L54: “regulating their expression”: more precise please, please explain the mechanism: mRNA stability/translation regulation

L55: misplaced comma?

L84: “RNase” instead of “RNAse”

L117: “It has been demonstrated a different expression of miRNAs associated with lipoprotein”, grammar/phrasing? please re-phrase

L120: “sick subjects” better: “patients” (?)

L123: “membrane-bound” the authors probably mean “membrane-enclosed”?

L124: “bounded” rather “bound”

L125: why only mention AGO2 here? It should be mentioned in the biogenesis part already! Almost all miRNAs are packaged into Ago2 during biogenesis.

L131: “RNase” instead of “RNAse”

L153: “expression” is not the correct term here since the authors refer to the patient serum. Rather “content” here

L172: “Circulating-miRNAs panel have been observed” not clear what is meant here.

L177: “serum five-miRNA panel” better “a panel of five miRNAs in serum” in case the authors meant that?

L179: The authors suddenly use the “-3p” and “-5p” nomenclature for miRNAs. The reader may not know what this means. Maybe it would be good to already introduce these terms when pre-miRNA strand selection is explained?

L214/215: suddenly italics font?

L248/249 better “it is” not “it’s”

L261: “circulating miRNAs” here? The authors are discussing miRNA in urine. Or do they mean that the same miRNAs are also circulating at the same time in blood? Please clarify and/or rephrase the text.

L288: “The analysis of Kaplan-Meier showed” please rephrase, maybe “Using Kaplan-Meier analysis” or “The analysis using Kaplan-Meier estimators” or related.

L312: “was correlated” passive

 L332: “miR” please use “miRNA” here. “miR” should only be used when naming a specific miRNA (with number)

L352: why not use “miRNA” (“microRNA” instead) here?

L365: analyzed

Table 2: “The AUC was...” “AUC” abbreviation not introduced

L389: “BRT” abbreviation not introduced

L414: better “chemotherapeutic treatment” then “drugs”

L436: “evidenced” better: “provided evidence that”

L448 delete “about”

L449: “differentially expressed” without “levels” or “differential expression levels”.
“differentially expressed levels” does not make sense.

L450/451: “The work has reported” please rephrase: “the work has shown” or “the scientists have reported” (?)

L454/455: “exosomal” do the authors really know that this miRNAs are packaged into exosomes?

L465/466: “Kalogirou et al. HAVE investigated”

L470: “acquire” is not the right term here. Possibly “appear to have” or something related.

L477: “in gene expression” (without “the”)

L484-488: Ref. 104-107, is there an overlap in miRNAs in these datasets?

L534/535 “enhancement” without “e” at the end

L546: delete “The” at the beginning of the sentence

L550-552: this “sponging” mechanism is not clear to the reviewer. please explain a bit more in detail.

L561/562: “Different stages of ccRCC have been proved in association with different expression profiles of miRNAs” “proven to be associated”?  Is that what the authors mean? Rephrase please.

L563: better: “have been found to be altered”

L564: “an enzyme acting the addition of” grammar (?). Please rephrase

L596: typo “rhe” -> “the”

L599: “immunotherapy-based therapy” rephrase, 2 times “therapy” uneccessary.

L599/600: “The miR-497-5p is able to decrease the number of clones in vitro and to foster apoptosis in renal cell carcinoma” -> since the experimental setup is not explained, “number of clones” does not tell the reader anything here. Please rephrase or elaborate.

L628: “organ failure” without “ ’s ”

L636: word order, better: “(key ribonuclease responsible for”

L643-645: word order, better: “The molecular stability of circulating miRNAs in liquid biopsy as plasma, serum and urine provides a great potential…”

Author Response

Dear Prof. Alesse Guest Editor (special issue “The Role of MicroRNA in Cancer”),

thanks for all useful comments raised by Reviewers.

We are able now to submit a revised version of our manuscript (ID: genes-1288870).

Please, find below a point-by-point response.

In order to speed up the revision process, we have highlighted our corrections in yellow on the manuscript.

We hope in a favourable outcome.

Sincerely,

Prof. Antonella Calogero on behalf of all Authors

Sapienza University of Rome

Faculty of Pharmacy and Medicine

Department of Medical Surgical Sciences and Biotechnologies

C.so della Repubblica 79 04100 Latina

antonella.calogero@uniroma1.it

Tel. +39-07731757234

REVIEWER 2

Comments and Suggestions for Authors: The authors provide a good and comprehensive overview on the literature on circulating miRNAs and their connection to cancer and specifically RCC. There is not much to add to this part, but the introduction, and especially the part on miRNA biogenesis has to be re-written. Please find specific comments below.  Although most of the text is written in good and clear English, the manuscript would somehow benefit by minor language editing possibly by an English native speaker. Some phrasing/word order does not sound right. Although, the reviewer has to admit not to be an English native speaker, so besides some corrections suggested below, probably has not found everything language/grammar-related. General notes/comments/corrections:

COMMENT 41: „non-coding” is found with or without the “minus” several times in the text

RESPONSE 41: The text has been edited, accordingly.

COMMENT 42: The authors sometimes use capital and sometimes small “M” when starting a sentence with “miRNA”. Both is fine, but please be consistent.

RESPONSE 42: We have used the M capital at the beginning of the sentences.

COMMENT 43: There are several double spaces in the text, please use full text search to remove.

RESPONSE 43: The text has been edited, accordingly

COMMENT 44: Specific corrections: Line (“L”) 31: delete “they”

RESPONSE 44: The text has been edited, accordingly

 COMMENT 45: L39: the reviewer is unsure if miRNAs can be termed “epigenetic”

RESPONSE 45: We specified that “MiRNAs play a role in epigenetic mechanisms”.

COMMENT 46: L42: better “approximately 170 nt”

RESPONSE 46: The text has been edited, accordingly

COMMENT 47: L46: asterisk should be explained

RESPONSE 47: The star has been removed.

COMMENT 48: L47: “most stable single strand“: too imprecisely written, please refer here for a better phrasing: https://onlinelibrary.wiley.com/doi/10.1002/wrna.1627

RESPONSE 48: we have better rephrased as suggested.

COMMENT 49: L47: The Ago protein has to be mentioned already in this section!

RESPONSE 49: The text has been re-phrased, accordingly

COMMENT 50: L53: to the reviewers knowledge miRNA binding to the 5’-UTR is rare, and some do bind to coding sequence. Please rephrase and use more recent literature to cite here!

RESPONSE 50: we have rephrased and add more recent references.

COMMENT 51: L54: “regulating their expression”: more precise please, please explain the mechanism: mRNA stability/translation regulation

RESPONSE 51: The text has been re-phrased, accordingly

COMMENT 52: L55: misplaced comma?

RESPONSE 52: yes, it was a misplaced comma

 COMMENT 53: L84: “RNase” instead of “RNAse”

RESPONSE 53: The text has been edited, accordingly

COMMENT 54: L117: “It has been demonstrated a different expression of miRNAs associated with lipoprotein”, grammar/phrasing? please re-phrase

RESPONSE 54: The text has been re-phrased, accordingly

 COMMENT 55: L120: “sick subjects” better: “patients” (?)

RESPONSE 55: The text has been edited, accordingly

 COMMENT 56: L123: “membrane-bound” the authors probably mean “membrane-enclosed”?

RESPONSE 56: We do. The text has been edited, accordingly

COMMENT 57: L124: “bounded” rather “bound”

RESPONSE 57: The text has been edited, accordingly

COMMENT 58: L125: why only mention AGO2 here? It should be mentioned in the biogenesis part already! Almost all miRNAs are packaged into Ago2 during biogenesis.

RESPONSE 58: The text has been re-phrased, accordingly

 COMMENT 59: L131: “RNase” instead of “RNAse”

RESPONSE 59: The text has been edited, accordingly

COMMENT 60: L153: “expression” is not the correct term here since the authors refer to the patient serum. Rather “content” here

RESPONSE 60: The text has been edited, accordingly

COMMENT 61: L172: “Circulating-miRNAs panel have been observed” not clear what is meant here.

RESPONSE 61: The text has been re-phrased, accordingly 

COMMENT 62: L177: “serum five-miRNA panel” better “a panel of five miRNAs in serum” in case the authors meant that?

RESPONSE 62: yes, we do. The text has been edited, accordingly

COMMENT 63: L179: The authors suddenly use the “-3p” and “-5p” nomenclature for miRNAs. The reader may not know what this means. Maybe it would be good to already introduce these terms when pre-miRNA strand selection is explained?

RESPONSE 63: The text has been re-phrased, accordingly 

COMMENT 64: L214/215: suddenly italics font?

RESPONSE 64: we edited the italics font

COMMENT 65: L248/249 better “it is” not “it’s”

RESPONSE 65: The text has been edited, accordingly

COMMENT 66: L261: “circulating miRNAs” here? The authors are discussing miRNA in urine. Or do they mean that the same miRNAs are also circulating at the same time in blood? Please clarify and/or rephrase the text.

RESPONSE 66: The text has been re-phrased, accordingly 

 COMMENT 67: L288: “The analysis of Kaplan-Meier showed” please rephrase, maybe “Using Kaplan-Meier analysis” or “The analysis using Kaplan-Meier estimators” or related.

RESPONSE 67: The text has been edited, accordingly

COMMENT 68: L312: “was correlated” passive

RESPONSE 68: The text has been edited, accordingly

COMMENT 69: L332: “miR” please use “miRNA” here. “miR” should only be used when naming a specific miRNA (with number)

RESPONSE 69: The text has been edited, accordingly

COMMENT 70: L352: why not use “miRNA” (“microRNA” instead) here?

RESPONSE 70: The text has been edited, accordingly

COMMENT 71: L365: analyzed

RESPONSE 71:  The text has been edited, accordingly

COMMENT 72: Table 2: “The AUC was...” “AUC” abbreviation not introduced

RESPONSE 72: we added the abbreviation at the end of the Table.

 COMMENT 73: L389: “BRT” abbreviation not introduced

RESPONSE 73: we used the whole term without abbreviation

 COMMENT 74: L414: better “chemotherapeutic treatment” then “drugs”

RESPONSE 74: The text has been edited, accordingly

 COMMENT 75: L436: “evidenced” better: “provided evidence that”

RESPONSE 75: The text has been edited, accordingly

COMMENT 76: L448 delete “about”

RESPONSE 76: The text has been edited, accordingly

COMMENT 77: L449: “differentially expressed” without “levels” or “differential expression levels”.
“differentially expressed levels” does not make sense.

RESPONSE 77: The text has been edited, accordingly

COMMENT 78: L450/451: “The work has reported” please rephrase: “the work has shown” or “the scientists have reported” (?)

RESPONSE 78:  The text has been edited, accordingly

COMMENT 79: L454/455: “exosomal” do the authors really know that this miRNAs are packaged into exosomes?

RESPONSE 79: the article that we mentioned, reports the presence of miRNAs as cargo in exosomes. Generally, exosomes are largely acknowledged as carrier of protein, cytokines, DNA as well as miRNAs.

COMMENT 80: L465/466: “Kalogirou et al. HAVE investigated”

RESPONSE 80: The text has been edited, accordingly

COMMENT 81: L470: “acquire” is not the right term here. Possibly “appear to have” or something related.

RESPONSE 81: The text has been edited, accordingly

COMMENT 82: L477: “in gene expression” (without “the”)

RESPONSE 82: The text has been edited, accordingly

 COMMENT 83: L484-488: Ref. 104-107, is there an overlap in miRNAs in these datasets?

RESPONSE 83: yes, we have summarized these articles as they cite the same miRNAs.

COMMENT 84: L534/535 “enhancement” without “e” at the end

RESPONSE 84: The text has been edited, accordingly

COMMENT 85: L546: delete “The” at the beginning of the sentence

RESPONSE 85: The text has been edited, accordingly

COMMENT 86: L550-552: this “sponging” mechanism is not clear to the reviewer. please explain a bit more in detail.

RESPONSE 86: It is an inhibitory mechanism, so the verb to sequester has been used instead of sponging.

COMMENT 87: L561/562: “Different stages of ccRCC have been proved in association with different expression profiles of miRNAs” “proven to be associated”?  Is that what the authors mean? Rephrase please.

RESPONSE 87: we have rephrased the sentence

COMMENT 88: L563: better: “have been found to be altered”

RESPONSE 88: The text has been edited, accordingly

COMMENT 89: L564: “an enzyme acting the addition of” grammar (?). Please rephrase

RESPONSE 89:  we have rephrased the sentence

COMMENT 90: L596: typo “rhe” -> “the”

RESPONSE 90: The text has been edited, accordingly

COMMENT 91: L599: “immunotherapy-based therapy” rephrase, 2 times “therapy” uneccessary.

RESPONSE 91: The text has been edited, accordingly

COMMENT 92: L599/600: “The miR-497-5p is able to decrease the number of clones in vitro and to foster apoptosis in renal cell carcinoma” -> since the experimental setup is not explained, “number of clones” does not tell the reader anything here. Please rephrase or elaborate.

RESPONSE 92: we meant the clonogenic capacity, so we have rephrased the sentence.

COMMENT 93: L628: “organ failure” without “ ’s ”

RESPONSE 92: The text has been edited, accordingly

COMMENT 93: L636: word order, better: “(key ribonuclease responsible for”

RESPONSE 93: The text has been edited, accordingly

COMMENT 94: L643-645: word order, better: “The molecular stability of circulating miRNAs in liquid biopsy as plasma, serum and urine provides a great potential…”

RESPONSE 94: The text has been edited, accordingly
